# Genome-Wide DNA Methylation and Gene Expression in Patients with Indolent Systemic Mastocytosis

**DOI:** 10.3390/ijms241813910

**Published:** 2023-09-10

**Authors:** Aleksandra Górska, Maria Urbanowicz, Łukasz Grochowalski, Michał Seweryn, Marta Sobalska-Kwapis, Tomasz Wojdacz, Magdalena Lange, Marta Gruchała-Niedoszytko, Justyna Jarczak, Dominik Strapagiel, Magdalena Górska-Ponikowska, Iwona Pelikant-Małecka, Leszek Kalinowski, Bogusław Nedoszytko, Danuta Gutowska-Owsiak, Marek Niedoszytko

**Affiliations:** 1Department of Allergology, Medical University of Gdansk, 7 Dębinki Street, 80-210 Gdansk, Poland; mnied@gumed.edu.pl; 2Biobank Lab, Department of Oncobiology and Epigenetics, University of Lodz, 90-237 Lodz, Polandmichal.seweryn@biol.uni.lodz.pl (M.S.); marta.sobalska@biol.uni.lodz.pl (M.S.-K.); dominik.strapagiel@biol.uni.lodz.pl (D.S.); 3Independent Clinical Epigenetics Laboratory, Pomeranian Medical University, 71-281 Szczecin, Poland; tomwoj@pum.edu.pl; 4Department of Dermatology, Venereology and Allergology, Medical University of Gdansk, 80-210 Gdansk, Poland; magdalena.lange@gumed.edu.pl (M.L.); boguslaw.nedoszytko@gumed.edu.pl (B.N.); 5Department of Clinical Nutrition, Medical University of Gdansk, 80-210 Gdansk, Poland; mg@gumed.edu.pl; 6Department of Medical Chemistry, Medical University of Gdansk, 80-210 Gdansk, Poland; magdalena.gorska-ponikowska@gumed.edu.pl; 7Department of Medical Laboratory Diagnostics–Biobank Fahrenheit, Medical University of Gdansk, 80-210 Gdansk, Poland; iwona.pelikant-malecka@gumed.edu.pl (I.P.-M.); leszek.kalinowski@gumed.edu.pl (L.K.); 8BioTechMed Centre, Department of Mechanics of Materials and Structures, Gdansk University of Technology, 80-233 Gdansk, Poland; 9Invicta Fertility and Reproductive Center, Molecular Laboratory, 81-740 Sopot, Poland; 10Laboratory of Experimental and Translational Immunology, University of Gdansk, Intercollegiate Faculty of Biotechnology, University of Gdansk and Medical University of Gdansk, 80-307 Gdansk, Poland; danuta.gutowska-owsiak@ug.edu.pl

**Keywords:** genome-wide DNA methylation, epigenetics, gene expression, mastocytosis

## Abstract

Mastocytosis is a clinically heterogenous, usually acquired disease of the mast cells with a survival time that depends on the time of onset. It ranges from skin-limited to systemic disease, including indolent and more aggressive variants. The presence of the oncogenic KIT p. D816V gene somatic mutation is a crucial element in the pathogenesis. However, further epigenetic regulation may also affect the expression of genes that are relevant to the pathology. Epigenetic alterations are responsible for regulating the expression of genes that do not modify the DNA sequence. In general, it is accepted that DNA methylation inhibits the binding of transcription factors, thereby down-regulating gene expression. However, so far, little is known about the epigenetic factors leading to the clinical onset of mastocytosis. Therefore, it is essential to identify possible epigenetic predictors, indicators of disease progression, and their link to the clinical picture to establish appropriate management and a therapeutic strategy. The aim of this study was to analyze genome-wide methylation profiles to identify differentially methylated regions (DMRs) in patients with mastocytosis compared to healthy individuals, as well as the genes located in those regulatory regions. Genome-wide DNA methylation profiling was performed in peripheral blood collected from 80 adult patients with indolent systemic mastocytosis (ISM), the most prevalent subvariant of mastocytosis, and 40 healthy adult volunteers. A total of 117 DNA samples met the criteria for the bisulfide conversion step and microarray analysis. Genome-wide DNA methylation analysis was performed using a MethylationEPIC BeadChip kit. Further analysis was focused on the genomic regions rather than individual CpG sites. Co-methylated regions (CMRs) were assigned via the CoMeBack method. To identify DMRs between the groups, a linear regression model with age as the covariate on CMRs was performed using Limma. Using the available data for cases only, an association analysis was performed between methylation status and tryptase levels, as well as the context of allergy, and anaphylaxis. KEGG pathway mapping was used to identify genes differentially expressed in anaphylaxis. Based on the DNA methylation results, the expression of 18 genes was then analyzed via real-time PCR in 20 patients with mastocytosis and 20 healthy adults. A comparison of the genome-wide DNA methylation profile between the mastocytosis patients and healthy controls revealed significant differences in the methylation levels of 85 selected CMRs. Among those, the most intriguing CMRs are 31 genes located within the regulatory regions. In addition, among the 10 CMRs located in the promoter regions, 4 and 6 regions were found to be either hypo- or hypermethylated, respectively. Importantly, three oncogenes—*FOXQ1*, *TWIST1*, and *ERG*—were identified as differentially methylated in mastocytosis patients, for the first time. Functional annotation revealed the most important biological processes in which the differentially methylated genes were involved as transcription, multicellular development, and signal transduction. The biological process related to histone H2A monoubiquitination (GO:0035518) was found to be enriched in association with higher tryptase levels, which may be associated with more aberrant mast cells and, therefore, more atypical mast cell disease. The signal in the BAIAP2 gene was detected in the context of anaphylaxis, but no significant differential methylation was found in the context of allergy. Furthermore, increased expression of genes encoding integral membrane components (*GRM2* and *KRTCAP3*) was found in mastocytosis patients. This study confirms that patients with mastocytosis differ significantly in terms of methylation levels in selected CMRs of genes involved in specific molecular processes. The results of gene expression profiling indicate the increased expression of genes belonging to the integral component of the membrane in mastocytosis patients (*GRM2* and *KRTCAP3*). Further work is warranted, especially in relation to the disease subvariants, to identify links between the methylation status and the symptoms and novel therapeutic targets.

## 1. Introduction

Systemic mastocytosis is a neoplastic hematological disease characterized by an expansion and accumulation of atypical mast cells (MCs) in multiple body organs, including bone marrow, skin, liver, spleen, lymph nodes, and the gastrointestinal tract. The clinical course of the disease depends on the WHO subvariant diagnosis [1] and systemic symptoms resulting from the release of the MC mediators, including anaphylaxis [2]. The prognosis of mastocytosis depends on the subvariant of the disease, from favorable in cutaneous mastocytosis (CM), not affecting life expectancy and characterized by a low progression rate in the indolent systemic mastocytosis (ISM) subvariant, to rapidly deteriorating aggressive forms of the disease, i.e., the aggressive systemic mastocytosis (ASM) subvariant and mast cell leukemia (MCL). The prognosis may change with disease progression or the effects of treatment. Recently, multiple therapeutic targets have been reported in neoplastic MCs, and several of those are currently the focus of clinical trials [1]. The crucial element in pathogenesis is the presence of the oncogenic somatic mutation in the *KIT* gene. Specifically, more than 90% of patients with systemic mastocytosis (SM) have a gain-of-function (GOF) mutation in the receptor tyrosine kinase [3]. In addition, further epigenetic alterations may have a compounding effect on the expression of genes beyond the modification of the DNA sequence. In general, it is accepted that DNA methylation inhibits the binding of transcription factors, thereby down-regulating gene expression; however, some transcription factors have been shown to bind to methylated sites [4]. DNA methylation, which is the most commonly known epigenetic change, predominantly occurs in the cytosines that precede guanines; these are called dinucleotide CpGs and are frequently observed in a variety of biological and pathological processes [5]. DNA methyltransferases (DNMTs) are involved in DNA methylation by catalyzing the transfer of a methyl group to the 5-position of the cytosine in DNA; this generates 5-methylcytosine (5mC) [6]. During the process of DNA demethylation, 5mC is catalyzed into 5-hydroxymethylcytosine (5hmC) by the TET hydroxylases, which play a key role in active DNA demethylation. Global DNA hypomethylation results in chromosome instability and leads to an increased incidence of cancer [7].

Changes in DNA methylation in the methylome (i.e., the genome-wide methylation profile) often affect gene expression with a specific functional result [8]. The effect of DNA methylation on gene expression links the methylation of CpG sites in the gene promoter region with a decrease in gene expression. To identify the functional regulatory role of DNA methylation, an understanding of its effect on gene expression is crucial [9]; bioinformatics tools have been proposed to enable the assessment of DNA methylation at CpG sites. It is clear that the regulating impact on gene expression depends on the genomic location of the DNA methylation [10,11]. Specific alterations to DNA methylation were shown previously to be associated with altered gene expression in the development of cancer and cardiovascular diseases [12]. Several epigenetic changes were described as potentially relevant to mastocytosis, including mutations in genes involved in epigenetic processes, such as TET2, DNMT3A, and ASXL1, and global and gene-specific methylation patterns.

A precise predictive tool for the prognosis of ISM patients and for decision making in individual management and treatment are highly active topics of research. Decreased DNA demethylation in the blood DNA of ISM patients recently indicated the involvement of epigenetic alterations in the pathology of mastocytosis [13]. Interestingly, the study also implied a possible role of allergic processes as an important epigenetic modifier and indicated the impairment of mast cell function in ISM patients without allergy symptoms [13].

Therefore, in order to explore the prevalence of epigenetic modifications in mastocytosis, we analyzed the methylomes of ISM patients to identify the differentially methylated regions (DMRs) and affected genes. This may open new treatment methods based on specific oncogene targets.

## 2. Results

Genome-wide methylation analysis was performed on DNA samples isolated from all blood cells in whole blood. For downstream analysis, data obtained from 712,665 probes and 103 individuals were used out of the 865,918 probes and 117 individuals present in the array at the beginning. The omitted samples comprised five samples removed for qualitative reasons, four outliers, four samples from patients treated with methotrexate or corticosteroids (Methylprednisolone or Encorton), and one from an individual diagnosed with breast cancer. The distributions of probes on chromosomes, in relation to the CpG sites and in relation to gene regions, are presented in Figure 1a–c.

The singular value distribution (SVD) analysis determined that differences in the blood cell composition and technical factors were the components that contributed the most to the variability in the studied dataset (Figure 2a).

As there were significant differences in blood cell composition between the groups, as presented in Table 1, a stepwise regression analysis was performed, considering the covariates and taking into account the enrichment of NK cells and granulocytes. Consequently, only these cells were corrected for cell composition. After adjustments were applied, the singular value decomposition (SVD) was repeated to ensure no variation due to confounding factors (Figure 2b).

The methylation variation analysis identified multiple DMRs between groups. The age covariant was added to the linear regression model, as the age differences between the study groups were statistically significant (*p* = 0.0003 in the Wilcoxon test).

A total of 85 CMRs, differing in terms of methylation levels between the groups, were also detected (statistically significant regions, assuming a Δβ cutoff level of >0.05); from those, 38 CMRs were identified as hypomethylated, and 47 CMRs were identified as hypermethylated.

Ten significant CMRs with annotations (regions with statistically significant differences between the groups and with Δβ cutoff > 0.05) are presented in Table 2; eight CMRs were found to be hypermethylated, and two were found to be hypomethylated.

Taking into account statistically significant differences between the groups (cut-off threshold Δβ > 0.05), we identified 10 CMRs located in the promoter regions (Table 3); 4 of those regions were found to be hypomethylated (*APOB*, *RASGEF1B*, *KRTCAP3*, and *FAM123C*), and 6 were found to be hypermethylated (*SH3PXD2A*, *SGMS*, *CDH*, *ADGRG6*, *PAQR7*, and *LOC100507195*).

All 85 CMRs (all selected CMRs with statistically significant differences between the groups and with a Δβ cutoff of >0.05) are described in the table in Appendix A. From these 85 CMRs, 38 CMRs were identified as hypomethylated (with reduced methylation), and 47 CMRs were identified as hypermethylated (with increased methylation). Among the 85 CMRs, 31 genes located in the regulatory regions were identified: *APOB*, *RASGEF1B*, *KRTCAP3*, *FAM123C*, *ANKMY1*, *GRM2*, *MFSD11*, *SLC2A14*, *PLSCR2*, *CCER2*, *RAB22A*, *SLC6A16*, *SYCP2L*, *NKAIN3*, *ANO1*, *SGPP2*, *TM1*, *SGPP2 EDARADD*, *CMC1*, *CCDC102B*, *SCG2*, *NEK6*, *TMEM246-AS1*, *TMEM220*, *PAQR7*, *ADGRG6*, *CDH7*, *SGMS1*, and *SH3PXD2A.*

Based on these 85 significant regions, three visualizations were made to illustrate their resolving power between the studied groups. Figure 3 shows the result of a principal component analysis (PCA) with a clear separation between the study groups. This difference is mainly visible on the first component, which explains 34.6% of the variation.

Based on the hierarchical clustering, a dendrogram was created, as shown in Figure 4. The algorithm determined two main groups; the first group includes all patients with mastocytosis, and the second includes healthy individuals from the control group.

A heat map of the beta values, standardized for individual CMRs for all tested individuals and the changes in methylation between mastocytosis patients and healthy controls, is shown in Figure 5.

**Figure 5 ijms-24-13910-f005:**
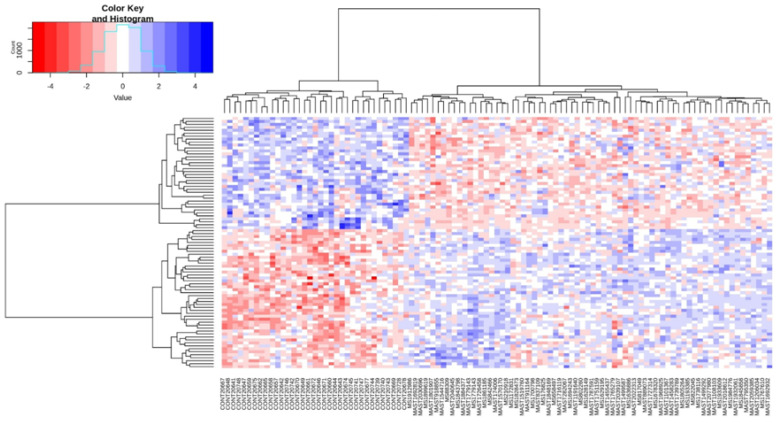
Heatmap of methylation levels of CpGs in the 85 differential CMRs identified in mastocytosis patients; comparison: mastocytosis vs. healthy. Blue colors indicate hypermethylated regions, and red colors indicate hypomethylated regions. Functional annotation distinguished 6 clusters in which differentially methylated regions of genes were observed, i.e., ‘cell adhesion via plasma membrane’, ‘multicellular organism development’, ‘transmembrane transport’, ‘signal peptide’, ‘transmembrane region and integral component of membrane’, and ‘cell junction’ (details provided in Table 4).

**Table 4 ijms-24-13910-t004:** Functional annotation of differentially methylated genes.

Genes	%	Pop Total	*p*-Value	Annotation Cluster Category
6 genes *PCDHGA4*, *CHL1*, *PCDHGB4*, *HEPACAM*, *LAMA1*, *CDH7*	8.571428571428571	20,581	0.01761274931909916	UP_KEYWORDS Cell adhesion
6 genes *EDARADD*, *SLC2A14*, *ANO1*, *SFRP1*, *HOXD13*, *ERG*	8.571428571428571	16,792	0.02384219242460433	GO:0007275~multicellular organism development
5 genes *ANO1*, *KCNJ9*, *GABRA5*, *FAM26F*, *KCNK2*	7.142857142857142	20,581	0.02680651305675513	UP_KEYWORDS Ion channel
21 genes *SLC2A14*, *PCDHGA4*, *PCDHGB4*, *HEPACAM*, *LAMA1*, *GABRA5*, *LGI2*, *NPY1R*, *SEZ6*, *SORCS1*, *MFSD11*, *PDGFRL*, *CDH7*, *GRM2*, *ANO1*, *SFRP1*, *CHL1*, *COL4A1*, *APOB*, *KCNK2*, *CTSC*	30.0	20,063	0.03726772532691608	UP_SEQ_FEATURE glycosylation site:N-linked (GlcNAc)
23 genes *SLC2A14*, *PCDHGA4*, *PCDHGB4*, *HEPACAM*, *KCNJ9*, *GABRA5*, *RNF180*, *NPY1R*, *SEZ6*, *FAM189A1*, *SORCS1*, *MFSD11*, *SGPP2*, *TMEM220*, *CDH7*, *GRM2*, *ANO1*, *KRTCAP3*, *NKAIN3*, *CHL1*, *PLSCR2*, *FAM26F*, *KCNK2*	32.857142857142854	20,063	0.06400605439867665	UP_SEQ_FEATURE transmembrane region
24 genes *SLC2A14*, *PCDHGA4*, *PCDHGB4*, *HEPACAM*, *GABRA5*, *RNF180*, *NPY1R*, *SEZ6*, *FAM189A1*, *SORCS1*, *MFSD11*, *SGPP2*, *TMEM220*, *CDH7*, *GRM2*, *ANO1*, *KRTCAP3*, *SFRP1*, *NKAIN3*, *CHL1*, *PLSCR2*, *FAM26F*, *GLCCI1*, *KCNK2*	34.285714285714285	18,224	0.07143941043520041	GO:0016021~integral component of membrane
4 genes *GRM2*, *CPEB1*, *GABRA5*, *DDB2*	5.714285714285714	18,224	0.19860048273016015	GO:0030054~cell junction

The most important biological processes in which the differentially methylated genes are involved are transcription, multicellular development, and signal transduction (Table 5), while the most important cellular components are from the category integral component of membrane (Table 6). Moreover, three of the identified genes, i.e., *FOXQ1*, *TWIST1*, and *ERG*, are linked with molecular functions—they have an impact on the activity of RNA polymerase II transcription factor (details provided in Table 7).

Interestingly, the most significant result of the gene disease association dataset analysis revealed 22 genes involved in the tobacco use disorder category (*MYO10*, *LAMA1*, *RNF180*, *NEK6*, *CRMP1*, *FAM189A1*, *SORCS1*, *MCC*, *PTPN13*, *CCDC102B*, *CDH7*, *RCAN1*, *NKAIN3*, *CHL1*, *COL4A1*, *ASXL3*, *TNIP3*, *ERG*, *APOB*, *KCNK2*, *L3MBTL4*, and *SH3GL2*) and 6 genes related to diseases linked with alterations in the lipid profile (cholesterol, HDL, and triglycerides). OMIM disease analysis also revealed two genes—*MCC* and *PDGFRL*—with an impact on the colorectal cancer process.

### 2.1. Differential Methylation Analysis in the Context of Tryptase Levels, Allergy, and Anaphylaxis

In the studied group, allergy was diagnosed in 62% of the mastocytosis patients, including to insect venom (50% of cases), drugs (30.4% of cases), food allergens (8.7% of cases), and pollens (8.7% of cases), while anaphylaxis had occurred in 43.2% of cases and was most commonly triggered by insect stings (72% cases) and drug intake (25% cases).

Using the available data for cases only (n = 67), we performed an association analysis between the methylation status and the tryptase levels (we assumed a tryptase level of 70 ng/mL as high). We utilized the linear model approach with the batch-effect-corrected data, as described above. To this end, we identified 746 probes significantly associated with tryptase levels, among which 377 were negatively associated and the remaining 369 were positively associated with the traits of interest. Additionally, we performed GO enrichment analysis; this identified one biological process, ‘GO:0035518–histone H2A monoubiquitination’, which was significantly (q < 0.05) associated with the tryptase levels.

Next, we aimed to detect differentially methylated probes in the context of allergy and anaphylaxis using case-only data; however, we were unable to detect any significantly differential signals at the level of the entire methylome. Hence, we proceeded with a targeted analysis, focusing on the genes previously identified to be differentially expressed in anaphylaxis [14]. We mapped the KEGG pathways, which were enriched due to the small sample size; we used the FDR correction within each of the pathways separately. Here, we detected a differential signal in only one gene, BAIAP2, associated with the 04810 pathway (“Regulation of actin cytoskeleton”), but found no differential methylation in the context of allergy overall.

### 2.2. Gene Expression Profiling

Twenty genes were selected for expression analysis using the RT-PCR method based on the CMRs located in the promoter regions. Significant differences in the expression of the selected genes were found for *GRM2* (*p* = 0.013) and *KRTCAP3* (*p* = 0.036) in mastocytosis patients compared to the heathy controls (Appendix A).

## 3. Discussion

While mastocytosis is a debilitating disease with a great impact on life expectancy and quality of life, the mechanism underlying the clinical symptoms of mastocytosis and disease progression remains unclear [15,16]. In this study, we investigated patients’ methylomes to gain insights into disease pathogenesis and to search for markers relevant to disease progression, beyond the KIT mutation status. The presence of oncogenic KIT mutation D816V has been detected in more than 80% of adult SM cases [17]. Other less common (<5%) somatic KIT mutations identified in adult SM include V560G, D815K, D816Y, insVI815-816, D816F, D816H, and D820G [15]. Activating KIT mutations are associated with mastocytosis, but it remains currently undetermined whether individual mutations are both necessary and sufficient to cause aberrant mast cells and the various clinical manifestations of mastocytosis. It is documented that DNA methylation affects gene expression [5,18,19]. Specifically, in cancer, DNA methylation plays an important role by regulating the expression of oncogenes, and the role of DNA methylation in the onset and progression of various neoplasm diseases has been presented [20]. The identification of the methylation patterns that affect gene expression may also be an important in mastocytosis and could potentially offer predictive value as a prognostic factor, with a consequent impact on individual therapy and disease management.

In our previous study, we demonstrated decreased DNA demethylation in the blood DNA of ISM patients [13]. We found a significantly lower content of the marker of hydroxymethylation (5-hmC) in the blood DNA of patients with mastocytosis compared to healthy individuals. We also identified a trend towards lower levels of the marker of methylation (5mC) and significantly higher levels of 5-hmC in patients with allergic symptoms as compared to patients without allergies. This suggests that allergic manifestations may be associated with DNA demethylation, leading to hypomethylation in mastocytosis or at least epigenetic changes in this disease [13].

Methylation variation analysis in the current study identified differentially methylated regions (DMRs) and established co-methylated regions (CMRs) differing in terms of methylation levels between the compared groups. From 85 CMRs, 38 CMRs were identified as hypomethylated, and 47 CMRs were identified as hypermethylated. Among these 85 CMRs, 31 genes located in the regulatory regions were identified (Appendix A). Among them, the most intriguing are the genes *KRTCAP3*, *ANKMY1*, and *GRM2*. The disease-affected genes we identified are predominantly involved in the formation of cellular components, signal transduction, multicellular organism development processes, and positive regulation of gene expression. We also found significant changes (hyper- and hypomethylation) involving the promoter regions: four regions were hypomethylated (*APOB*, *RASGEF1B*, *KRTCAP3*, and *FAM123C* genes) and six were hypermethylated (*SH3PXD2A*, *SGMS*, *CDH*, *ADGRG6*, *PAQR7*, and *LOC100507195* genes).

The identification of three oncogenes differentially methylated in mastocytosis patients, i.e., *FOXQ1, pro*, and *ERG*, was an interesting finding. Aberrant expression of *FOXQ1* and *TWIST1* has been previously noted in epithelial–mesenchymal transition (EMT), contributing to metastasis in several tumors [21]. Specifically, FOXQ1 has been observed to increase metastatic competence and drug resistance through triggering EMT in carcinoma cells [22]. *FOXQ1* expression is predictive of a worse prognosis in six different solid tumors and may be explored as a potential therapeutic target in the future [23]. Similarly, *TWIST1* has a pronounced effect on cancer progression by enhancing EMT, stimulating the proliferation and invasiveness of cancer cells, and promoting metastasis and resistance to chemotherapy [24]. Finally, *ERG* has been identified as being consistently overexpressed in malignant epithelial cells in prostate cancer and has also been shown to be involved in proliferation, as well as vasculo- and angiogenesis. Overexpression of *ERG* has been observed in several cancers, e.g., leukemia and Ewing’s sarcoma [25], and was also shown in approximately half of all prostate cancer patients. The most probable mechanism that has been proposed for the role of *ERG* in cancer is linked to the formation of gene fusion of *TMPRSS2* and *ERG* [20]. These results suggest that epigenetic changes in oncogenes may trigger the pathogenesis of mastocytosis and may lead to more aberrant mast cells and, therefore, more advanced disease.

The analysis of gene expression confirmed upregulation of the genes associated with integral components of the cell membrane, i.e., *GRM2* and *KRTCAP3*, previously identified by us as being potentially important in the disease through epigenetic microarrays. Notably, regulatory regions within these genes, including the hypomethylated promoter region of *KRTCAP3*, were detected. Interestingly, both genes seem to also be important in the development of neoplastic diseases, but no link has been demonstrated so far for mastocytosis.

GRM2 (glutamate receptor, metabotropic 2) is part of group 2 of the metabotropic glutamate receptors (mGluRs), belonging to a family of G-protein-coupled receptors which participate in the modulation of synaptic transmission and neuronal excitability throughout the central nervous system [26]. The mGluRs bind glutamate within a large extracellular domain and transmit signals to intracellular signaling partners [26]. GRM2 is involved in processes linked to the inhibition of the cyclic AMP cascade. The widespread expression of mGluRs makes these receptors particularly attractive drug targets, and recent studies continue to validate the therapeutic utility of mGluR ligands in neurological and psychiatric disorders such as Alzheimer’s disease, Parkinson’s disease, anxiety, depression, and schizophrenia [26]. Patients with mastocytosis have a significantly lower quality of life and experience increased anxiety, depression, and other symptoms such as headaches and fatigue [27,28]. Therefore, the increased expression of the GRM2 gene could be investigated for a possible link to neurological and psychiatric symptoms in mastocytosis.

KRTCAP3 (keratinocyte-associated protein 3) is preferentially expressed in cultured primary human keratinocytes compared to dermal fibroblasts. Based on the available data (www.gtexportal.org, accessed on 1 January 2023), KRTCAP3 is highly expressed in the skin and also in the digestive tract, the male and female sex organs, the pituitary and thyroid glands, the lungs, and the pancreas. The protein is predicted to be an integral membrane protein; it has previously been identified as a candidate adiposity gene in human and rat GWAS studies [29]. Studies on KRTCAP3 in rats showed its role in food consumption and insulin sensitivity, with important sex differences [29]. Decreased expression of KRTCAP3 led to weight gain in female rats on a high-fat diet, while in male rats, it led to an increase in insulin resistance without weight gain.

In the context of cancer, hypermethylated KRTCAP3 was identified in melanoma samples and melanoma cell lines compared to normal melanocytes, while no clear difference in mRNA expression was found [4,23].

Alterations in the methylome have also been shown for osteosarcoma (OS) [30]. Moreover, this study showed that the dysregulation of specific methylated genes was correlated with the metastasis-free time in patients with OS; these included ANKMY1 and KRTCAP3, which were also identified in our project. Interestingly, it was shown that higher expression of KRTCAP3 was associated with longer metastasis-free survival time in OS patients [30]. Thus, the hypomethylation and increased expression of KRTCAP3, which we found in mastocytosis, could be potentially linked to a better prognosis; this requires further studies in the context of progression to the advanced forms of the disease.

In our previous study, we found a tendency towards reduced levels of demethylation markers and, simultaneously, significantly higher levels of hydroxymethylation in patients with allergic symptoms compared to patients without allergies [13]. However, in our current study, we failed to detect differentially methylated genes in the context of allergy using case-only data. Further analyses carried out on a selected set of genes previously described in anaphylaxis [14] detected only one potentially interesting hit, i.e., the *BAIAP2* gene, annotated to the pathway of regulation of the actin cytoskeleton; there have been no published data describing such a link so far. The most prevalent allergic reaction in mastocytosis patients is anaphylaxis to *Hymenoptera* venom. We plan further studies to compare patients with solely *Hymenoptera* allergy and a group with concomitant mastocytosis, secondly to confirm the results in larger populations.

In conclusion, our study established that patients with mastocytosis demonstrate alterations within their methylome, which could be further explored mechanistically and as prognostic markers. Further work is warranted, especially in relation to the disease subvariants, to identify links between the methylation status and the symptoms. Further studies on the relevance of oncogenes *FOXQ1*, *TWIST1*, and *ERG* as second-hit mutations in mastocytosis and novel therapeutic targets are needed.

## 4. Methods and Materials

### 4.1. Case–Control Study

#### Population

The study population comprised 80 ISM patients included in the local registry and treated at the Allergology Department, Medical University of Gdansk, between 2012 and 2020. Mastocytosis was diagnosed according to the WHO guidelines [1], which included a pathological examination of bone marrow aspirate (cytological evaluation, mast cell immunophenotyping with assessment of CD2 and CD25 expression), identification of the presence of the activating point mutation in *KIT*, and measurement of the serum tryptase level [2]. Molecular analysis of the c.2447A > T variant (KIT p. Asp816Val) in the *KIT* gene was performed via qPCR in bone marrow aspirate [31]. Our local registry is part of the European Competence Network on Mastocytosis (ECNM) Registry [2]. The gene expression analysis was performed in a random group of 20 cases who were not included in the epigenetic study. The control group included 40 healthy adult volunteers, matched for sex and age and free of chronic diseases, including allergy. Peripheral blood samples were collected from ISM patients at the point of diagnosis or during a follow-up visit at the Allergology Department and from healthy volunteers recruited by the Biobank Lab, University of Lodz. Informed consent was obtained from all the study participants. The database of the ECNM registry, data storage, and data distribution comply with the rules and regulations of data protection laws, with the respective local ethical committee regulations for each participating center, and with the Declaration of Helsinki. The enrolment for the study was performed between January and December 2019. The study was approved by the Independent Bioethics Committee for Scientific Research at the Medical University of Gdańsk: NKBBN/270/2018.

### 4.2. Methods

#### 4.2.1. Genome-Wide Methylation Profiling

DNA was isolated from 200 µL of whole blood using a MagNA Pure LC DNA Isolation Kit and MagNA Pure LC 2.0 Instrument (Roche, Basel, Switzerland), according to the manufacturer’s protocol. DNA was quantified using a broad range Quant-iT™ dsDNA Broad Range Assay Kit (Invitrogen™, Carlsbad, CA, USA). All the DNA samples underwent quality control using a PCR reaction for sex determination. For all of the samples for which the DNA concentration was above 40 ng/uL, and for which the sex determined by PCR was consistent with that described in the questionnaire were enrolled to the further procedures. A total of 117 DNA samples met the criteria for the bisulfide conversion step and further microarray analysis. A quantity of 500 ng DNA from each sample was independently treated with sodium bisulfite using an EZ DNA Methylation Kit™ (Zymo Research, Irvine, CA, USA). A genome-wide DNA methylation analysis was performed using a recently developed MethylationEPIC BeadChip (Illumina, San Diego, CA, USA), which covers more than 850,000 CpG sites. All procedures were conducted according to the manufacturers’ instructions.

#### 4.2.2. Gene Expression Using Real-Time PCR

Gene expression profiling was conducted in a group of patients diagnosed with mastocytosis (n = 20) and a group of healthy controls (n = 20). Total RNA was isolated from whole blood using a Magna Pure LC RNA Isolation Kit on a Magna Pure device (Roche, Basel, Switzerland) according to the manufacturer’s instructions. The isolation was supported by a DNAse digestion step to remove genomic DNA. The isolated RNA underwent qualitative and quantitative assessment using a NanoDrop spectrophotometer (NanoDrop, USA) and a Bioanalyzer 2100 (Agilent Technologies, Perlan, Les Ulis, France). Following this, 500 ng of RNA was reverse-transcribed into cDNA using a SuperScript IV Reverse Transcriptase Kit (ThermoFisher Scientific, Waltham, MA, USA). Real-time PCR analysis was performed with the use of TaqMan Probes (ThermoFisher Scientific, Waltham, MA, USA) in a BioRad ThermoCycler CFX (Bio-Rad, Hercules, CA, USA). Reactions were carried out in duplicate with a non-template control on 384-well optical reaction plates. The expression was assessed for the following genes: *ABCA2* (ATP-Binding Cassette Subfamily A Member 2), *CDH7* (cadherin 7), *DNMT3A* (DNA Methyltransferase 3 Alpha), *EAPP* (E2F-associated phosphoprotein), *EDARADO* (EDAR-associated death domain), *GRM2* (glutamate metabotropic receptor 2), *HDAC9* (Histone Deacetylase 9), *KRTCAP3* (keratinocyte-associated protein 3), *OTX2* (Orthodenticle Homeobox 2), *RAB22A* (RAB22A, member of the RAS oncogene family), *RASGEF* (RasGEF domain family member 1B), *RUNX1* (Runt-related transcription factor 1), *SCG2* (secretogranin II), *SETD2* (SET Domain Containing 2), *SGMS1* (sphingomyelin synthase 1), *SH3PXD2* (SH3 and PX domains 2A), *SLC2A14* (solute carrier family 2 member 14), *TET2* (Tet Methylcytosine Dioxygenase 2), and *TPSB2* (Tryptase Beta 2). The obtained gene expression profiles were normalized to that of PPIB as the reference gene. All data, including the raw cycle threshold (Ct), were used for the comparative Ct method [32]. All variables were tested for the normality of their distribution, and relative gene expressions were transformed to a natural logarithm scale.

### 4.3. Bioinformatic and Statistical Analysis

#### Analysis of the Genome-Wide Methylation

Raw fluorescent intensity data (.idat files) were analyzed using Chip Analysis of the Methylation Pipeline (ChAMP) [33]. In the initial filtering, probes that failed to attain a *p*-value sufficient for detection (cut-off of ≤0.01), were represented <3 times in 5% of samples, were non-CpG probes, were on sex chromosomes, were related with polymorphic sites [34], or were non-specific were filtered out [35]. Samples with less than 10% of detected probes were filtered out, and four additional samples were removed for being outliers, as indicated by principal component analysis (PCA) on the 1000 most variable probes. For downstream analysis, data from 712,665 probes and 103 individuals were used.

Data were normalized using the beta-mixture quantile (BMIQ) method [36] to correct for type 1 and type 2 bias. Cell decomposition revealed significant differences in the enrichment of NK cells, monocytes, and granulocytes between the studied groups using three different methods: the Houseman method implemented in RefBaseEWAS [37,38], Robust Partial Correlations (RPC) [39], and Cibersort (CBS) [40], both implemented in EpiDISH [41] (Appendix A). The influence of the cell type was removed using RefBaseEWAS. Singular value distribution (SVD) [42] was used to estimate the influence of each available variable. Batch effect removal was carried out using ComBat [43].

Further analysis was focused on the genomic regions rather than individual CpG sites, since regulatory DNA modifications generally involve multiple consecutive CpGs. Co-methylated regions (CMRs) were assigned via the CoMeBack method [44], which combines sites into regional units to reflect the biology of DNA methylation and is independent of any variables of interest. To identify DMRs between groups, a linear regression model with age as the covariate on CMRs was performed using Limma [45]. For DMRs to be considered significant, mean β values had to have a differential of at least 5% (|Δβ| > 0.05) between groups with *p*-values of ≤9 × 10^−8^, as recommended by Mansell et al. [46].

Using the available data for mastocytosis cases only (n = 67), we also performed an association analysis between methylation status and levels of tryptase (we assumed a tryptase level of 70 ng/mL as high). We utilized the linear model approach with the batch-effect-corrected data, as described above. In what follows, we aimed to detect differentially methylated regions in the context of allergy and anaphylaxis using case-only data. In both cases, we were unable to detect signals at the methylome-wide level of significance (i.e., with q values below 0.05). Therefore, we analyzed the genes that were described previously as being differentially expressed in anaphylaxis, as reported by Niedoszytko et al. [14]. Due to the small sample size, we used FDR correction within each of the pathways separately.

Finally, we also performed a gene ontology (GO) enrichment analysis using + DAVID software (https://david.ncifcrf.gov, accessed on 2 November 2020) based on genes in which CMRs were found.

## Figures and Tables

**Figure 1 ijms-24-13910-f001:**
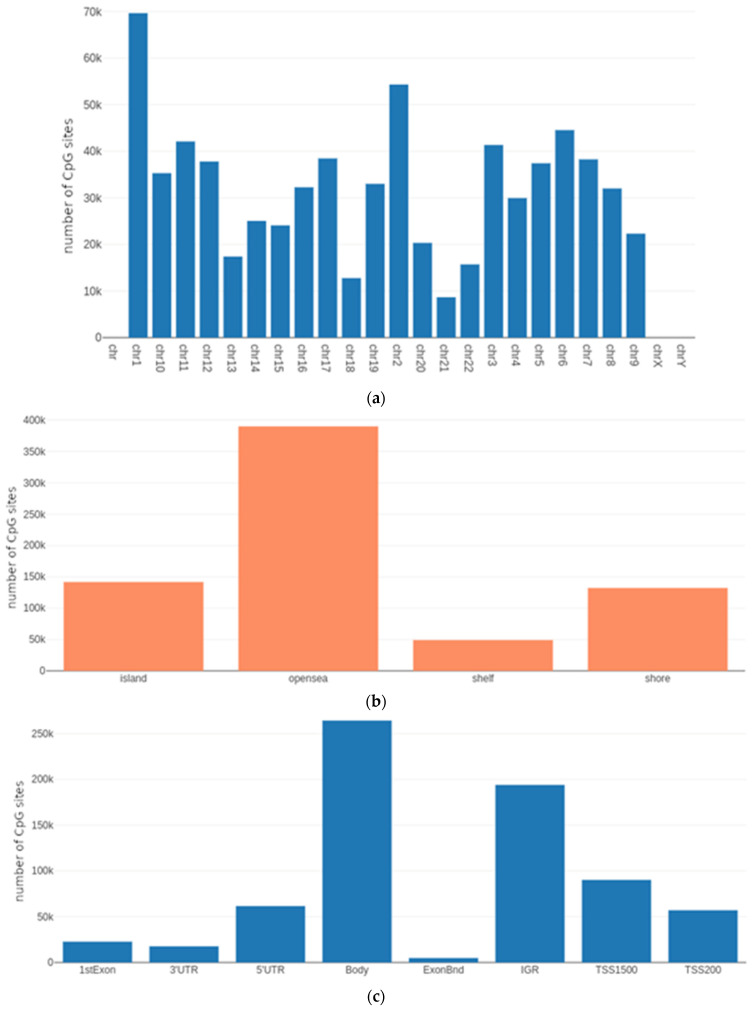
(**a**) The chromosomal distribution of probes. (**b**) Probe distribution in relation to the CpG islands. Island: a region with an increased number of methylated CpG sites next to each other; Open sea: a region not related to the CpG island (distance > 4 kb); Shelf: a region in the genome 2–4 kbp from the CpG island; Shore: a region within 2 kbp of the CpG island. (**c**) Probe distribution in relation to the location within the gene. 1stExon: the region located in the first exon; 3′UTR: untranslated region located in the 3′ direction from the coding sequence; 5′UTR: untranslated region located in the 5′ direction from the coding sequence; Body: the region of the gene that includes all exons and introns; ExonBnd: a region located on the boundaries of an exon; IGR: intergenic region; TSS1500: site distant from the start of transcription by 200 to 1500 nucleotides; TSS200: site less than 200 nucleotides from the start of transcription.

**Figure 2 ijms-24-13910-f002:**
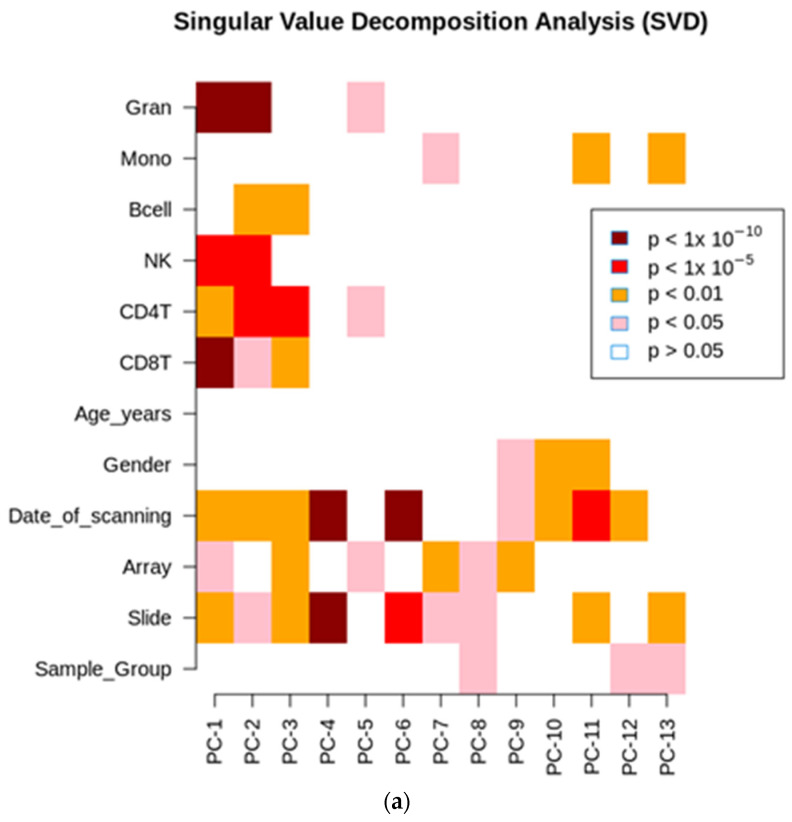
(**a**) Variability within the individual components of the unadjusted SVD analysis. (**b**) Variability within the individual components of the adjusted SVD analysis.

**Figure 3 ijms-24-13910-f003:**
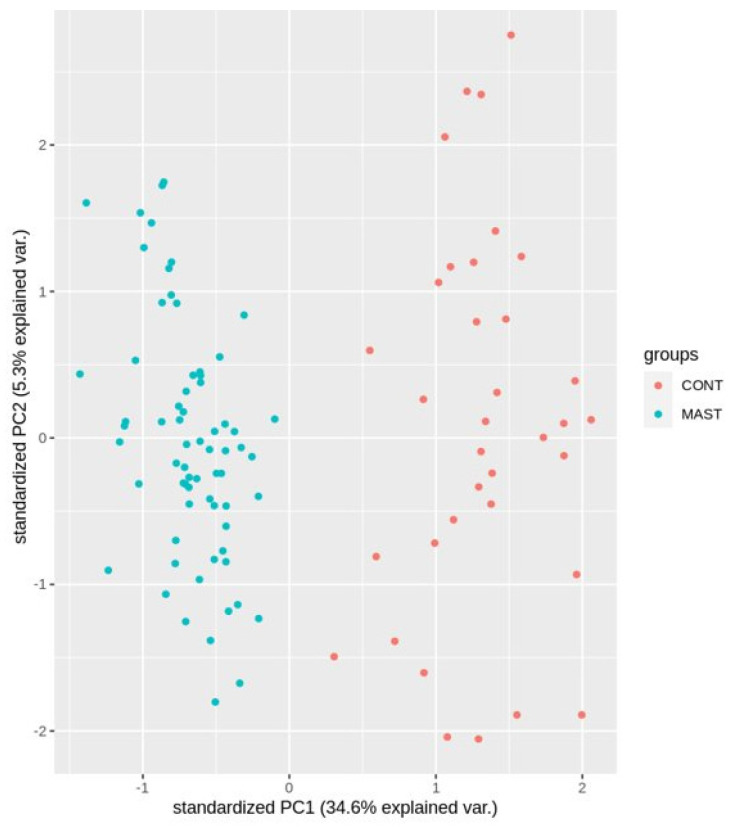
PCA of the 85 designated CMRs.

**Figure 4 ijms-24-13910-f004:**
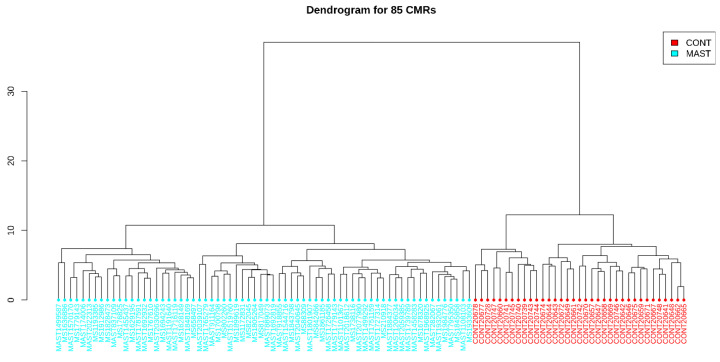
Dendrogram of the 85 CMRs that differentiate patients with mastocytosis from healthy controls.

**Table 1 ijms-24-13910-t001:** *p*-values from the Wilcoxon test for cell proportions, as determined by three algorithms (statistically significant values are marked in red).

Cells Proportions	Wilcoxon *p*-Value Houseman	Wilcoxon *p*-Value RPC	Wilcoxon *p*-Value CBS
CD8T	0.0532	0.0629	0.0707
CD4T	0.9196	0.9639	0.9362
NK	0.0053	0.0038	0.0029
Bcell	0.7833	0.9917	0.7779
Mono	0.0063	0.0037	0.0045
Gran	0.0088	0.0101	0.0103

**Table 2 ijms-24-13910-t002:** Summary of the 10 most variable CMRs.

chr	Start	End	Gene	Feature	no.cpgs	deltaBeta	*p*-Value	CMR	cgi
chr19	12876846	12877188	*HOOK2*	Body	4	0.3328	7.8249 × 10^−8^	CMR1	shore
chr1	225924665	225924683		IGR	2	0.2376	2.2321 × 10^−17^	CMR2	opensea
chr3	16924563	16924709		IGR	2	0.1515	5.5525 × 10^−12^	CMR3	shore
chr2	29179066	29179435		IGR	2	0.1456	1.4603 × 10^−9^	CMR4	opensea
chr22	49447845	49448320		IGR	3	−0.1378	6.1083 × 10^−8^	CMR5	island
chr1	212148758	212149423	*INTS7*	Body	2	0.1222	1.2099 × 10^−17^	CMR6	opensea
chr10	105616094	105616523	*SH3PXD2A*	TSS1500	2	0.1217	2.5599 × 10^−10^	CMR7	shore
chr22	37493737	37494173	*TMPRSS6*	Body	2	0.1087	2.5023 × 10^−12^	CMR8	island
chr10	52134715	52135449	*SGMS1*	5′UTR	2	0.1047	1.4676 × 10^−11^	CMR9	opensea
chr2	21267858	21268152	*APOB*	TSS1500	2	−0.1008	1.4746 × 10^−10^	CMR10	shore

Table 2 and Table 3: deltaBeta: difference in the average level of methylation between the mastocytosis group and the control group; *p*-Value: *p*-value for each CMR; CMR: name of the CMR; gene: region in which the given CMR is located; feature: the CMR’s position in relation to the gene; cgi: location of the CMR in relation to the CpG islands; chr: chromosome; start: CMR starting position on the chromosome; stop: CMR ending position on the chromosome; no.cpgs: number of CpG sites in a given CMR present on the EPIC matrix.

**Table 3 ijms-24-13910-t003:** Summary of the 10 most variable CMR promoter regions.

chr	Start	End	Gene	Feature	no.cpgs	deltaBeta	*p*-Value	CMR	cgi
chr10	105616094	105616523	*SH3PXD2A*	TSS1500	2	0.1217	2.5599 × 10^−10^	CMR7	shore
chr10	52134715	52135449	*SGMS1*	5′UTR	2	0.1047	1.4676 × 10^−11^	CMR9	opensea
chr2	21267858	21268152	*APOB*	TSS1500	2	−0.1008	1.4746 × 10^−10^	CMR10	shore
chr4	82392459	82392533	*RASGEF1B*	5′UTR	2	−0.0997	2.3214 × 10^−8^	CMR11	shore
chr18	63416728	63417139	*CDH7*	TSS1500	3	0.0957	3.0711 × 10^−9^	CMR13	shore
chr6	142621875	142622515	*ADGRG6*	TSS1500	4	0.0880	3.5338 × 10^−11^	CMR16	shore
chr2	27664918	27666036	*KRTCAP3*	TSS1500	11	−0.0858	1.5291 × 10^−8^	CMR18	shore
chr1	26198721	26199190	*PAQR7*	TSS1500	2	0.0798	8.5844 × 10^−9^	CMR19	shelf
chr2	131512651	131512663	*FAM123C*	TSS1500	2	−0.0737	5.2954 × 10^−10^	CMR26	shore
chr12	68845785	68845935	*LOC100507195*	TSS1500	2	0.0729	1.3801 × 10^−8^	CMR27	opensea

**Table 5 ijms-24-13910-t005:** The most important biological processes in which the differentially methylated genes were involved.

Biological Process	Genes	%	*p*-Value	Pop Total
GO:0071456~cellular response to hypoxia	*SFRP1, CPEB1, TWIST1, KCNK2*	5.714285714285714	0.0034491518494985792	16,792
GO:0060527~prostate epithelial cord arborization involved in prostate glandular acinus morphogenesis	*SFRP1, HOXD13*	2.857142857142857	0.01879156240102788	16,792
GO:0060687~regulation of branching involved in prostate gland morphogenesis	*SFRP1, HOXD13*	2.857142857142857	0.02188962311858626	16,792
GO:0007275~multicellular organism development	*EDARADD, SLC2A14, ANO1, SFRP1, HOXD13, ERG*	8.571428571428571	0.02384219242460433	16,792
GO:0007165~signal transduction	*EDARADD, RCAN1, MYO10, CHL1, GABRA5, NEK6, ERG, MCC, SH3GL2*	12.857142857142856	0.02814390092226652	16,792
GO:0010628~positive regulation of gene expression	*FUBP1, RNF207, TWIST1, APOB*	5.714285714285714	0.04971198712285179	16,792
GO:0090102~cochlea development	*GABRA5, KCNK2*	2.857142857142857	0.07311172522948037	16,792
GO:0007156~homophilic cell adhesion via plasma membrane adhesion molecules	*PCDHGA4, PCDHGB4, CDH7*	4.285714285714286	0.08883878862456275	16,792

**Table 6 ijms-24-13910-t006:** The most important cellular components in which the differentially methylated genes were involved.

Cellular Components	Genes	%	*p*-Value	Pop Total
GO:0016021~integral component of membrane	*SLC2A14*, *PCDHGA4*, *PCDHGB4*, *HEPACAM*, *GABRA5*, *RNF180*, *NPY1R*, *SEZ6*, *FAM189A1*, *SORCS1*, *MFSD11*, *SGPP2*, *TMEM220*, *CDH7*, *GRM2*, *ANO1*, *KRTCAP3*, *SFRP1*, *NKAIN3*, *CHL1*, *PLSCR2*, *FAM26F*, *GLCCI1*, *KCNK2*	34.285714285714285	0.07143941043520041	18,224
GO:0005886~plasma membrane	*MYO10*, *PCDHGA4*, *PCDHGB4*, *KCNJ9*, *GABRA5*, *NPY1R*, *SEZ6*, *MCC*, *PTPN13*, *RAB22A*, *CDH7*, *GRM2*, *ANO1*, *SFRP1*, *NKAIN3*, *CHL1*, *PLSCR2*, *APOB*, *KCNK2*, *SH3GL2*	28.57142857142857	0.07801374005885228	18,224
GO:0043025~neuronal cell body	*MYO10*, *SEZ6*, *APOB*, *KCNK2*	5.714285714285714	0.08865085033682481	18,224

**Table 7 ijms-24-13910-t007:** The most important molecular function in which the differentially methylated genes were involved.

Molecular Function	Genes	%	*p*-Value	Pop Total
GO:0000981~RNA polymerase II transcription factor activity, sequence-specific DNA binding	*FOXQ1*, *TWIST1*, *ERG*	4.285714285714286	0.09428895074022843	16,881

## Data Availability

Data is contained within the article or Appendix A.

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
