# Peer review of "Genome-Wide DNA Methylation and Gene Expression in Patients with Indolent Systemic Mastocytosis"

_ijms, 2023, doi:10.3390/ijms241813910_

Round 1

Reviewer 1 Report

Minor suggestions.  Line 31: The aim of the study "is" rather than was.  Line 49:  hypo-or-hypermethylation.  Line 50:  TWIST1, and ERG.  Line 103:  "Previously", rather than years ago.  Line 106:  "highly active topics of research", rather than hot topics. Line 132: "center" rather than centre which is the British spelling.  Line 154:  "whole blood", rather than the whole blood.  Line 179:  "were", rather than was. Line 215: "on DNA samples" rather than in DNA samples.  Figure 1.b  The Figure 1. b. the description of the figure components, e.g. starts with" Island", then it goes to "shore", then back to "Shelf", then back to "open sea" , is there a relevance to that?  "technical factors" vague ? suggest description of removal.  Figures 3,4,and 5 are not easy to see, however I believe these will be much clearer in the final print of the manuscript.  Line 395 "thein" ? , this may be obsolete.

Having said all that, I believe these are minor changes that can be done. Overall, this an excellent manuscript on Mastocytosis, addressing very important issues, as well as addressing recent issues. I believe that this is an important up-to-date information regarding Mastocytosis, and should be accepted for publication after minor changes.

I recommend changing centre to "center", and favourable to favorable. 

Author Response

Dear Sir,

Thank You for Your comments and suggestions.

Please see the attachment.Refeer

Reviewer 2 Report

Overall Feedback:

The manuscript attempts to elucidate the role of DNA methylation in systemic mastocytosis, which is commendable given the need to understand the molecular underpinnings of this condition. The paper is organized systematically and provides a plethora of information. However, there are several areas where clarity, consistency, and relevance can be improved. The provided comments and suggestions should guide authors to ensure that their work is presented in the most comprehensive and transparent manner.

Main Suggestions:

  1. Clarity and Conciseness: A recurring theme across the abstract, introduction, and discussion sections is the need for clarity and brevity. While detailed information is valuable, it's crucial to ensure that the manuscript remains concise, especially in the abstract and introduction.
  2. Methodological Rigor: The method section requires a bit more detail in certain areas, especially concerning sample sizes, selection criteria, and justification for specific techniques or thresholds. This will allow for a more transparent replication and evaluation of the study by peers.
  3. Relevance and Context: In both the results and discussion sections, while several genes and pathways are mentioned, the direct relevance or implications for systemic mastocytosis aren't always clear. Offering a more in-depth context can help readers grasp the importance of the findings.
  4. Avoiding Redundancy: Ensure that the same information isn't repeated in close proximity, as seen in some sections of the discussion. Repetition can dilute the manuscript's impact and reduce reader engagement.
  5. Linking the Study: It's vital to make clear links between this study's findings and prior research, especially if the authors' previous work is mentioned. This helps establish the progression of knowledge and highlights the novelty of the current study.

In conclusion, the manuscript presents promising data that could be pivotal in understanding systemic mastocytosis's molecular aspects. By addressing the above points and refining the presentation, the authors can ensure that their findings are both impactful and well-received by the scientific community.

Abstract

1.       Clarity and Structure: The abstract starts with a detailed background about mastocytosis. However, the transition between the background and the objective is a bit abrupt. The abstract could benefit from a more concise background and a smoother transition into the study's objectives.

  1. Sampling: The study population is limited to "adult patients with indolent systemic mastocytosis (ISM)". This limits the generalization of the findings to only this subtype of mastocytosis. It would be beneficial to clarify why this specific subgroup was chosen for study and if the results can be generalized to other forms of mastocytosis.
  2. Methodology: The bisulfite conversion step criteria, which led to the exclusion of some samples, is mentioned but not detailed. Clarification on why some samples didn't meet the criteria could be provided.
  3. Statistics and Analysis: It would be useful to know more about the statistical significance of the methylation differences observed. Were the p-values adjusted for multiple comparisons? How strong were the associations, and were effect sizes mentioned?
  4. Results Interpretation: The mention of "3 oncogenes – FOXQ1, TWIST1, ERG – were identified as differentially methylated in mastocytosis patients" is intriguing. However, the functional or clinical significance of this differential methylation is not clarified. Are these genes previously known to be associated with mastocytosis or other diseases?
  5. Consistency: The results section mentions significant methylation differences in 85 CMR regions. However, later details focus on only 31 genes within those regions and 10 CMRs in promoter areas. The abstract could benefit from more consistent representation of these numbers.
  6. Relevance: The significance of the biological process related to "histone H2A monoubiquitination (GO:0035518)" and its association with higher tryptase levels is not immediately clear. The authors might consider elaborating on the relevance of this finding in the context of mastocytosis.
  7. Conclusion: The conclusion could benefit from mentioning any potential clinical or therapeutic implications of the findings. Given the importance of the role of methylation in disease pathology, the abstract should elucidate if these findings have potential therapeutic implications or if they can be used as biomarkers for mastocytosis progression.

Introduction

1.       Clarity and Structure: The introduction appears lengthy and somewhat repetitive. The essence of systemic mastocytosis and the role of DNA methylation could be conveyed in a more concise manner.

  1. Citations: Line 66 makes reference to the WHO subvariant diagnosis but does not provide a specific reference ("...depends on the WHO subvariant diagnosis1..."). Please ensure that citations are correctly linked to the corresponding statement.
  2. Jargon and Technical Language: Lines 77-104 are heavy with biological jargon that may be challenging for readers unfamiliar with the topic. Consider simplifying some of the technical terms or providing a brief explanation where necessary.
  3. Specificity and Detail: The description of the KIT gene mutation is specific (i.e., where a valine is substituted for an aspartate) but might be overly detailed for an introduction, especially if it's not central to the paper's main focus.
  4. Flow of Ideas: The transition between the initial discussion on systemic mastocytosis and the detailed description of DNA methylation seems abrupt. Consider incorporating a bridging sentence to help guide the reader from one topic to the next.
  5. Redundancy: The idea that increased DNA methylation in the promoter region represses gene expression is reiterated in lines 94-95 and 99-100. Please avoid repeating the same point in such close succession.
  6. Ambiguity: In lines 100-101, the statement mentions a "positive association of DNA methylation to gene expression in prostate cancer". This could be clearer; as it stands, it's unclear how this is relevant to systemic mastocytosis. Ensure that all examples or analogies directly support the topic at hand.
  7. Consistency: The introduction starts with a detailed discussion on systemic mastocytosis and later shifts to DNA methylation. While it's clear by the end that the paper will be looking at the role of DNA methylation in systemic mastocytosis, it may be beneficial to introduce this link earlier on to maintain a consistent focus throughout the introduction.
  8. Concluding Statement: The concluding lines (110-112) effectively introduce the aim of the paper. However, ensure that the body of the paper directly addresses and delivers on this aim.
  9. Clinical Relevance: Consider emphasizing the clinical significance or potential implications of understanding DNA methylation patterns in systemic mastocytosis. This will make the research more compelling for a broader audience.

Method

Case-control study

  1. Sample size and justification: The sample size for both the case and control groups should be justified. Were power calculations performed? A 2:1 ratio of cases to controls is a bit unusual. Ideally, a 1:1 ratio is more common to maximize statistical power.
  2. Control selection: The control group, consisting of healthy adult volunteers, should provide more detail on the selection criteria. Were they matched only by age and sex? Were other potential confounding factors considered?
  3. Kit Mutation: The selection of the c.2447A> T variant (KIT p. Asp816Val) in the KIT gene for qPCR evaluation needs justification. Are there other variants in the KIT gene that might be clinically relevant?

2.2. Methods

  1. DNA isolation and quantification: You've mentioned using 200 µl of whole blood for DNA isolation, but it would be useful to have details about the initial concentrations of DNA before they underwent quality control.
  2. Quality control: The exact criteria for DNA quality control are not mentioned. How was it determined that 117 DNA samples met the criteria?
  3. Gene expression: For gene expression analysis, only a subset of patients was used (n=20 for both cases and controls). Why was this subset chosen? Is it a random subset? And how are the results of this subset generalizable to the entire patient population?

3. Bioinformatic and statistical analysis

  1. Probe filtering: The filtering steps are thorough, but it might be worth including justification for each filtering step, especially for the removal of non-CpG probes and those on sex chromosomes.
  2. Batch effect correction: The use of ComBat for batch effect removal is appropriate, but were potential confounding factors considered in this correction?
  3. Differential methylation analysis: The threshold of |Δβ| > 0.05 is set for DMRs, but what was the rationale behind this threshold? Additionally, why was the threshold of P values ≤ 9*10-8 chosen, as recommended by Mansell et al.29?
  4. Association analysis: The case-only analysis is interesting but may not have adequate power given the sample size. This may explain the absence of significant q-values. Were power calculations performed for this analysis?
  5. Gene ontology (GO) enrichment: The GO enrichment analysis was performed on genes with identified CMRs, but were these genes also found to be differentially methylated in your primary analysis?

Discussion

  1. Introductory statement clarity: The initial statement (lines 364-366) about mastocytosis being debilitating could benefit from a reference to provide evidence of this claim.
  2. Relevance of previous work: While the authors mentioned their previous study (lines 376-383), it would be beneficial to explain its significance and how it builds on or complements the current study.
  3. Precision of the terms used: In line 368, the term "beyond the KIT mutation status" is mentioned. A brief clarification or context about the relevance and importance of the KIT mutation in mastocytosis might help readers unfamiliar with the topic.
  4. Expression clarity: The statement "It is documented that DNA methylation affects gene expression" is somewhat vague (line 369). A more specific citation or reference would be beneficial here.
  5. Clarity in results presentation: In lines 384-391, it would be helpful if the authors provided more detailed explanations or insights on how these differentially methylated regions (DMRs) and co-methylated regions (CMRs) specifically relate to mastocytosis. The sheer listing of genes without deeper context may be overwhelming for the readers.
  6. Relevance of oncogenes: While the authors noted the identification of three oncogenes (lines 398-413), they should also discuss the potential importance or implications of these findings for mastocytosis specifically.
  7. Context for the gene description: The explanations provided for GRM2 (lines 421-434) and KRTCAP3 (lines 435-444) are detailed but could benefit from clearer elucidation on their roles or implications in mastocytosis.
  8. Potential redundancy: The authors repeated some of the information about their previous study (lines 456-461), which they had already mentioned earlier. Avoiding repetitions can make the text more concise and reader-friendly.
  9. Clarification needed: In lines 461-466, it's mentioned that there was a failure to detect differentially methylated genes in the context of allergy. It would be helpful to elaborate on possible reasons for this and its implications.
  10. Conclusion and future directions: The conclusion (lines 467-470) is appropriate, but the authors may want to suggest more specific next steps or methodologies for further work, rather than just a general call for further research.

Overall Feedback:

The manuscript attempts to elucidate the role of DNA methylation in systemic mastocytosis, which is commendable given the need to understand the molecular underpinnings of this condition. The paper is organized systematically and provides a plethora of information. However, there are several areas where clarity, consistency, and relevance can be improved. The provided comments and suggestions should guide authors to ensure that their work is presented in the most comprehensive and transparent manner.

Main Suggestions:

  1. Clarity and Conciseness: A recurring theme across the abstract, introduction, and discussion sections is the need for clarity and brevity. While detailed information is valuable, it's crucial to ensure that the manuscript remains concise, especially in the abstract and introduction.
  2. Methodological Rigor: The method section requires a bit more detail in certain areas, especially concerning sample sizes, selection criteria, and justification for specific techniques or thresholds. This will allow for a more transparent replication and evaluation of the study by peers.
  3. Relevance and Context: In both the results and discussion sections, while several genes and pathways are mentioned, the direct relevance or implications for systemic mastocytosis aren't always clear. Offering a more in-depth context can help readers grasp the importance of the findings.
  4. Avoiding Redundancy: Ensure that the same information isn't repeated in close proximity, as seen in some sections of the discussion. Repetition can dilute the manuscript's impact and reduce reader engagement.
  5. Linking the Study: It's vital to make clear links between this study's findings and prior research, especially if the authors' previous work is mentioned. This helps establish the progression of knowledge and highlights the novelty of the current study.

In conclusion, the manuscript presents promising data that could be pivotal in understanding systemic mastocytosis's molecular aspects. By addressing the above points and refining the presentation, the authors can ensure that their findings are both impactful and well-received by the scientific community.

Abstract

1.       Clarity and Structure: The abstract starts with a detailed background about mastocytosis. However, the transition between the background and the objective is a bit abrupt. The abstract could benefit from a more concise background and a smoother transition into the study's objectives.

  1. Sampling: The study population is limited to "adult patients with indolent systemic mastocytosis (ISM)". This limits the generalization of the findings to only this subtype of mastocytosis. It would be beneficial to clarify why this specific subgroup was chosen for study and if the results can be generalized to other forms of mastocytosis.
  2. Methodology: The bisulfite conversion step criteria, which led to the exclusion of some samples, is mentioned but not detailed. Clarification on why some samples didn't meet the criteria could be provided.
  3. Statistics and Analysis: It would be useful to know more about the statistical significance of the methylation differences observed. Were the p-values adjusted for multiple comparisons? How strong were the associations, and were effect sizes mentioned?
  4. Results Interpretation: The mention of "3 oncogenes – FOXQ1, TWIST1, ERG – were identified as differentially methylated in mastocytosis patients" is intriguing. However, the functional or clinical significance of this differential methylation is not clarified. Are these genes previously known to be associated with mastocytosis or other diseases?
  5. Consistency: The results section mentions significant methylation differences in 85 CMR regions. However, later details focus on only 31 genes within those regions and 10 CMRs in promoter areas. The abstract could benefit from more consistent representation of these numbers.
  6. Relevance: The significance of the biological process related to "histone H2A monoubiquitination (GO:0035518)" and its association with higher tryptase levels is not immediately clear. The authors might consider elaborating on the relevance of this finding in the context of mastocytosis.
  7. Conclusion: The conclusion could benefit from mentioning any potential clinical or therapeutic implications of the findings. Given the importance of the role of methylation in disease pathology, the abstract should elucidate if these findings have potential therapeutic implications or if they can be used as biomarkers for mastocytosis progression.

Introduction

1.       Clarity and Structure: The introduction appears lengthy and somewhat repetitive. The essence of systemic mastocytosis and the role of DNA methylation could be conveyed in a more concise manner.

  1. Citations: Line 66 makes reference to the WHO subvariant diagnosis but does not provide a specific reference ("...depends on the WHO subvariant diagnosis1..."). Please ensure that citations are correctly linked to the corresponding statement.
  2. Jargon and Technical Language: Lines 77-104 are heavy with biological jargon that may be challenging for readers unfamiliar with the topic. Consider simplifying some of the technical terms or providing a brief explanation where necessary.
  3. Specificity and Detail: The description of the KIT gene mutation is specific (i.e., where a valine is substituted for an aspartate) but might be overly detailed for an introduction, especially if it's not central to the paper's main focus.
  4. Flow of Ideas: The transition between the initial discussion on systemic mastocytosis and the detailed description of DNA methylation seems abrupt. Consider incorporating a bridging sentence to help guide the reader from one topic to the next.
  5. Redundancy: The idea that increased DNA methylation in the promoter region represses gene expression is reiterated in lines 94-95 and 99-100. Please avoid repeating the same point in such close succession.
  6. Ambiguity: In lines 100-101, the statement mentions a "positive association of DNA methylation to gene expression in prostate cancer". This could be clearer; as it stands, it's unclear how this is relevant to systemic mastocytosis. Ensure that all examples or analogies directly support the topic at hand.
  7. Consistency: The introduction starts with a detailed discussion on systemic mastocytosis and later shifts to DNA methylation. While it's clear by the end that the paper will be looking at the role of DNA methylation in systemic mastocytosis, it may be beneficial to introduce this link earlier on to maintain a consistent focus throughout the introduction.
  8. Concluding Statement: The concluding lines (110-112) effectively introduce the aim of the paper. However, ensure that the body of the paper directly addresses and delivers on this aim.
  9. Clinical Relevance: Consider emphasizing the clinical significance or potential implications of understanding DNA methylation patterns in systemic mastocytosis. This will make the research more compelling for a broader audience.

Method

Case-control study

  1. Sample size and justification: The sample size for both the case and control groups should be justified. Were power calculations performed? A 2:1 ratio of cases to controls is a bit unusual. Ideally, a 1:1 ratio is more common to maximize statistical power.
  2. Control selection: The control group, consisting of healthy adult volunteers, should provide more detail on the selection criteria. Were they matched only by age and sex? Were other potential confounding factors considered?
  3. Kit Mutation: The selection of the c.2447A> T variant (KIT p. Asp816Val) in the KIT gene for qPCR evaluation needs justification. Are there other variants in the KIT gene that might be clinically relevant?

2.2. Methods

  1. DNA isolation and quantification: You've mentioned using 200 µl of whole blood for DNA isolation, but it would be useful to have details about the initial concentrations of DNA before they underwent quality control.
  2. Quality control: The exact criteria for DNA quality control are not mentioned. How was it determined that 117 DNA samples met the criteria?
  3. Gene expression: For gene expression analysis, only a subset of patients was used (n=20 for both cases and controls). Why was this subset chosen? Is it a random subset? And how are the results of this subset generalizable to the entire patient population?

3. Bioinformatic and statistical analysis

  1. Probe filtering: The filtering steps are thorough, but it might be worth including justification for each filtering step, especially for the removal of non-CpG probes and those on sex chromosomes.
  2. Batch effect correction: The use of ComBat for batch effect removal is appropriate, but were potential confounding factors considered in this correction?
  3. Differential methylation analysis: The threshold of |Δβ| > 0.05 is set for DMRs, but what was the rationale behind this threshold? Additionally, why was the threshold of P values ≤ 9*10-8 chosen, as recommended by Mansell et al.29?
  4. Association analysis: The case-only analysis is interesting but may not have adequate power given the sample size. This may explain the absence of significant q-values. Were power calculations performed for this analysis?
  5. Gene ontology (GO) enrichment: The GO enrichment analysis was performed on genes with identified CMRs, but were these genes also found to be differentially methylated in your primary analysis?

Discussion

  1. Introductory statement clarity: The initial statement (lines 364-366) about mastocytosis being debilitating could benefit from a reference to provide evidence of this claim.
  2. Relevance of previous work: While the authors mentioned their previous study (lines 376-383), it would be beneficial to explain its significance and how it builds on or complements the current study.
  3. Precision of the terms used: In line 368, the term "beyond the KIT mutation status" is mentioned. A brief clarification or context about the relevance and importance of the KIT mutation in mastocytosis might help readers unfamiliar with the topic.
  4. Expression clarity: The statement "It is documented that DNA methylation affects gene expression" is somewhat vague (line 369). A more specific citation or reference would be beneficial here.
  5. Clarity in results presentation: In lines 384-391, it would be helpful if the authors provided more detailed explanations or insights on how these differentially methylated regions (DMRs) and co-methylated regions (CMRs) specifically relate to mastocytosis. The sheer listing of genes without deeper context may be overwhelming for the readers.
  6. Relevance of oncogenes: While the authors noted the identification of three oncogenes (lines 398-413), they should also discuss the potential importance or implications of these findings for mastocytosis specifically.
  7. Context for the gene description: The explanations provided for GRM2 (lines 421-434) and KRTCAP3 (lines 435-444) are detailed but could benefit from clearer elucidation on their roles or implications in mastocytosis.
  8. Potential redundancy: The authors repeated some of the information about their previous study (lines 456-461), which they had already mentioned earlier. Avoiding repetitions can make the text more concise and reader-friendly.
  9. Clarification needed: In lines 461-466, it's mentioned that there was a failure to detect differentially methylated genes in the context of allergy. It would be helpful to elaborate on possible reasons for this and its implications.
  10. Conclusion and future directions: The conclusion (lines 467-470) is appropriate, but the authors may want to suggest more specific next steps or methodologies for further work, rather than just a general call for further research.

Author Response

Dear Sir,

Thank You for Your comments and suggestions which improved the quality of the manuscript.

All revisions to the manuscript have been highlighted, such that any changes can be easily reviewed again by editors and reviewers. 

Please find attached file with the answer to Your questions, the description of the edition and corrections which have been made according to Your suggestions and comments.

Regards,

Aleksandra Górska
